# Prognostic difference between surgery and external radiation in patients with stage I liver cancer based on competitive risk model and conditional survival rate

**Rong Chen**[1]*, **Yanli An**[2], **Muhao Xu**[2]

1 Department of Oncology, Zhongda Hospital, Southeast University, Nanjing, Jiangsu Province, China,
2 Medical School of Southeast University, Nanjing, Jiangsu Province, China

* 101011581@seu.edu.cn

## Abstract

### Purpose

This study aimed to assess the difference in prognosis of patients with early-stage liver cancer after surgery or external radiation.

### Methods

Between 2010 and 2015, 2155 patients with AJCC 7[th] stage I liver cancer were enrolled in the SEER database. Among these, 1972 patients had undergone surgery and 183 had undergone external beam radiation. The main research endpoints were overall survival (OS) and disease-specific survival (DSS). The competitive risk model was used to calculate the risk ratio of liver cancer-specific deaths when there was a competitive risk. Propensity Score Matching (PSM) method using a 1:1 ratio was used to match confounders such as sex, age, and treatment method. Conditional survival was dynamically assessed for patient survival after surgery or external radiation.

### Results

Multivariate analysis of the competitive risk model showed that age, disease diagnosis time, grade, and treatment [surgery and external beam radiation therapy (EBRT)] were independent prognostic factors for patients with hepatocellular carcinoma. Surgery had a higher survival improvement rate than that of EBRT. As the survival of patients with liver cancer increased, the survival curve of surgery declined more slowly than that of radiotherapy patients and stabilized around 3 years after surgery. The survival curve of radiotherapy patients significantly dropped within 4 years and then stabilized.

### Conclusion

Surgery was better than EBRT for patients with stage I liver cancer. Close follow-up was required for 3 years after surgery or 4 years after external radiation. This study can help clinicians make better informed clinical decisions.

**Data Availability Statement:** All relevant data are within the manuscript and its Supporting Information files.

**Funding:** National Natural Science Foundation of China (81827805), National Key R&D Program of China (2018YFA0704100, 2018YFA0704104). The funders had no role in study design, data collection and analysis, decision to publish, or preparation of the manuscript.

**Competing interests:** The authors have declared that no competing interests exist.

## Introduction

Liver cancer is one of the most common malignant tumors in the world and has a high mortality rate [1]. Surgical resection is an important approach in patients with stage I liver cancer to achieve long-term survival. Since less than 30% of patients with early-stage liver cancer can tolerate radical surgery [2], researchers are actively exploring non-surgical interventions. The most commonly used treatment strategy for such patients is external beam radiation therapy (EBRT) [3].

A study based on the Surveillance, Epidemiology, and End Results (SEER) database, analyzed the 6th edition AJCC stage I hepatocellular carcinoma (HCC) patients who were diagnosed between 2004 and 2013. It is believed that for small tumors ($\leq$ 3 cm), the efficacy of EBRT is comparable to that of surgical resection [4]. With the development of technology in recent years, the indications of EBRT for liver cancer have expanded from adjuvant therapy to high-dose radical radiation therapy [5]. This research employed COX analysis, ignored the risk of competition, and may have misestimated the risk of the event of interest. For this reason, a competitive risk prognosis model based on liver cancer was further constructed and evaluated to provide reference for the selection of treatment alternatives and predict prognosis of patients with liver cancer.

Conditional survival rate is the probability of surviving for (x) years and then surviving for (y) years. As the survival time increases, the patients' risk of death dynamically changes [6]. Taking this into consideration, compared to the traditional cumulative survival rate, conditional survival rate can provide a more accurate and dynamic survival rate estimate. In addition, there are competing events of non-liver cancer associated deaths in the SEER database. If the competitive events exceed 10%, the use of the traditional COX proportional hazard model for overall survival (OS) analysis will overestimate the cumulative incidence of outcome events. In our study, a total of 523 people died in OS, of which 300 people died in DSS (Disease Specific Survival) (accounting for 57.36% of total deaths); thus, the proportion of deaths due to competitive events was 42.64% (100% - 57.36%).

Therefore, this study aimed to use the data of the 7th edition AJCC stage I liver cancer patients diagnosed between 2010 to 2015 from the SEER database, explore the prognostic value of two treatment modalities based on the competitive risk model. The propensity score matching (PSM) method was used to achieve the effect of "after the fact" randomization and conditional survival to analyze the conditional survival probability of the two treatment methods. To the best of our knowledge, there are no previous reports for the studied treatment modalities.

## Methods

### Data source

In this study, raw data are made available on SEER(https://seer.cancer.gov/) database of the United States National Cancer Institute. The data are public and do not involve the privacy of patients; therefore, review and approval by the ethics committee are not required. The SEER data are public and do not involve the privacy of patients, so the review and approval by the ethics committee are not required.

### Patient selection

Patients with stage I HCC diagnosed between 2010 and 2015 according to the 7th edition of AJCC were included. Inclusion criteria are (1) no distant metastasis, (2) single lesion, and (3) patients undergoing surgery or EBRT. Exclusion criteria are (1) patients with a history of malignant tumors and (2) patients with a survival period of $<$ 3 months.

## Endpoints

The primary endpoint was disease specific survival (DSS), which is defined as the time from surgery or EBRT till HCC-associated death. Non-liver cancer-associated death was a competitive event.

## Statistics

All statistical analyses were performed using R software (Version 4.0.5). Patient characteristics were compared by using Chi-square test for categorical data and Kruskal–Wallis test for continuous data. Survival analysis was performed using the Kaplan–Meier method for the estimation of the survival function. The log-rank test was used to compare the survival of patients according to the treatment modality (surgery versus EBRT). Cox regression analysis was used in OS and DSS single factor and multiple factor analysis. Cumulative incidence function was used to calculate the cumulative occurrence of liver cancer-specific deaths and non-specific deaths rates. Competitive risk models for single-factor and multi-factor analyses were used to calculate the risk ratio of liver cancer-specific death.

In the present study, the 1:1 PSM analysis (Caliper value setting: 0.1, method: nearest, R package: 'MatchIt' package) was performed by treatment. Matching factors included sex, age, disease stage, and other covariates. Conditional survival (CS) refers to the possibility of surviving for a certain number of years or months based on the survival for a certain period of time. It is a statistical method that reflects the dynamic characteristics of the prognosis after cancer diagnosis or treatment. A $p$-value $< 0.05$ was considered statistically significant and a two-sided test was performed.

## Results

### Baseline characteristics and survival analysis

Table 1 summarized the patients' baseline demographic characteristics. A total of 2155 patients with HCC met the inclusion criteria. Among them, 1,972 (91.51%) patients underwent surgical treatment to primary tumor site, while only 183 (8.5%) patients underwent EBRT.

The Kaplan–Meier plot was utilized to compare the OS and DSS. The DSS was the same for patients with or without chemotherapy, diagnosed at different times, or of different race and sex. As shown in Fig 1, patients <60 years, with grade I tumor, married or widowed, who underwent surgery, had longer OS and DSS than those >60 years, with a tumor grade III or IV, divorced or separated, who underwent EBRT. Patients who were diagnosed before 2011 had longer OS than those after 2014, but showed no difference in DSS. There were no differences in OS and DSS of patients with or without chemotherapy, irrespective of race and sex. Each variable fulfilled the PH (Proportion hypothesis) test (S1 and S2 Figs).

In multivariate Cox analysis (Fig 2), age, tumor grade, and surgery persevered to be independently associated with better OS (hazards ratio [HR] = 1.328, 95% confidence interval [CI] [1.109, 1.59], $p = 0.002$), (HR = 1.987, 95% CI [1.482, 2.665], $p < 0.001$), (HR = 0.195, 95% CI [0.15, 0.255], $p < 0.001$), respectively. These variables were also independently associated with better DSS (HR = 1.37, 95% CI [1.08, 1.739], $p = 0.01$), (HR = 2.453, 95% CI [1.671, 3.603], $p < 0.001$), (HR = 0.166, 95% CI [0.117, 0.234], $p < 0.001$), respectively, in matched population. The presence or absence of chemotherapy intake, time of diagnosis, marital status, race, and sex had no significant effect either on the OS or on the DSS.

Subgroup analysis and interaction test of COX proportional hazard model (Fig 3) shows that all patients can benefit from different treatments, and there is no difference between different groups, except for grade III and IV patients.

**Table 1. Demographic characteristics of the patients.**

| Variables | Total (n = 2155) | EBRT (n = 183) | Surgery (n = 1972) | p |
|---|---|---|---|---|
| Survival months, Median (Q1, Q3) | 32(19, 51) | 21 (14, 33) | 33 (20, 52.25) | <0.001 |
| OS, n (%) | | | | <0.001 |
| 0 | 1632 (76) | 71 (39) | 1561 (79) | |
| 1 | 523 (24) | 112 (61) | 411 (21) | |
| DSS, n (%) | | | | <0.001 |
| 0 | 1855 (86) | 113 (62) | 1742 (88) | |
| 1 | 300 (14) | 70 (38) | 230 (12) | |
| age, Median (Q1, Q3) | 61 (56, 68) | 64 (58, 72) | 61 (56, 67) | <0.001 |
| Sex, n (%) | | | | 0.884 |
| Female | 581 (27) | 48 (26) | 533 (27) | |
| Male | 1574 (73) | 135 (74) | 1439 (73) | |
| Race, n (%) | | | | <0.001 |
| Black | 251 (12) | 17 (9) | 234 (12) | |
| Others | 511 (24) | 18 (10) | 493 (25) | |
| White | 1393 (65) | 148 (81) | 1245 (63) | |
| Marital, n (%) | | | | 0.039 |
| Divorced/Separated | 261 (12) | 32 (17) | 229 (12) | |
| Married | 1269 (59) | 92 (50) | 1177 (60) | |
| Single/Unmarried | 392 (18) | 35 (19) | 357 (18) | |
| Widowed/Others | 233 (11) | 24 (13) | 209 (11) | |
| Diagnosis, n (%) | | | | 0.812 |
| 2012~2013 | 667 (31) | 53 (29) | 614 (31) | |
| Time< = 2011 | 631 (29) | 54 (30) | 577 (29) | |
| Time> = 2014 | 857 (40) | 76 (42) | 781 (40) | |
| Chemotherapy, n (%) | | | | <0.001 |
| NO | 1610 (75) | 90 (49) | 1520 (77) | |
| YES | 545 (25) | 93 (51) | 452 (23) | |
| Grade, n (%) | | | | <0.001 |
| I | 511 (24) | 18 (10) | 493 (25) | |
| II | 914 (42) | 17 (9) | 897 (45) | |
| III~IV | 272 (13) | 11 (6) | 261 (13) | |
| Unknown | 458 (21) | 137 (75) | 321 (16) | |
| Age. cat, n (%) | | | | 0.022 |
| ~60 | 1004 (47) | 70 (38) | 934 (47) | |
| 60~ | 1151 (53) | 113 (62) | 1038 (53) | |

## Competitive risk model

Cumulative risk curve in Fig 4 ('Cmprsk' package using R language) showed that the cumulative incidence was lower in the group <60 years ($p$ <0.05) and underwent surgery than that in the group >60 years and underwent EBRT ($p$ <0.001). There was no statistical difference of cumulative incidence in groups diagnosed at different times, of different race and sex, or with or without chemotherapy.

The variables of the competitive risk model were further studied by multivariate analysis (Fig 5). In summary, age >60 years vs. <60 years (HR = 1.241, 95% CI [1.002,1.537], $p$ = 0.048), diagnosis time during or after 2014 vs. between 2012 and 2013 (HR = 0.719, 95% CI [0.533,0.97], $p$ = 0.031), grade III and IV vs. grade I (HR = 2.168, 95% CI [1.544,3.045],

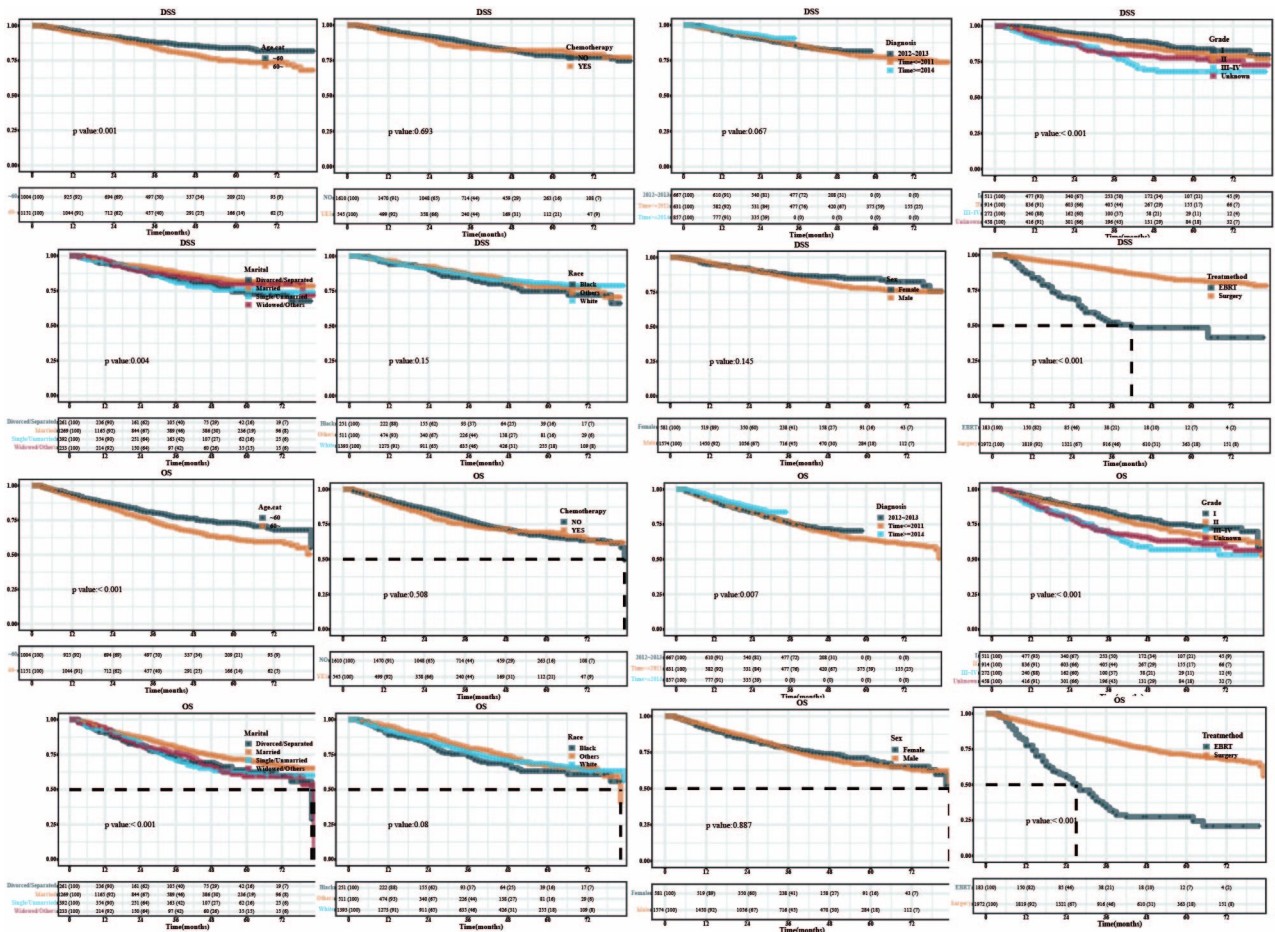

**Fig 1. KM curve of DSS & OS.**

$p < 0.001$), and surgery vs. EBRT (HR = 0.206, 95% CI [0.149,0.284], $p <$0.001) respectively, presented as independent factors of HCC patients.

Subgroup analysis and interaction test of competitive risk model in Fig 6 shows that all patients can benefit from different treatments, and there is no difference between different groups, except for grade III and IV patients.

## Propensity score matching

The PSM method was subsequently used to balance the baseline characteristics of the two groups of patients who underwent surgery and EBRT. After 1:1 PSM was used, all variables were well-balanced between the two groups of patients (Table 2). Density (before/after) and Hist (before/after) was used to test the balance of baseline data before and after PSM (S3 Fig).

In the competing risk model, for both univariate and multivariate analysis, we found that surgery had a higher rate of survival improvement than EBRT (HR = 0.224, 95% CI, [0.139, 0.36], $p <$0.001) (Fig 7). However, age, sex, race, marital status, time of diagnosis, chemotherapy intake, and tumor grade were not risk factors for patient survival (all $p >$0.05).

After PSM, subgroup analysis and interaction test of competitive risk model (Fig 8) showed that patients of different ages, sexes, races, marital status, time of diagnosis, and whether receiving chemotherapy or not can all benefit from surgery and EBRT. The treatment benefit

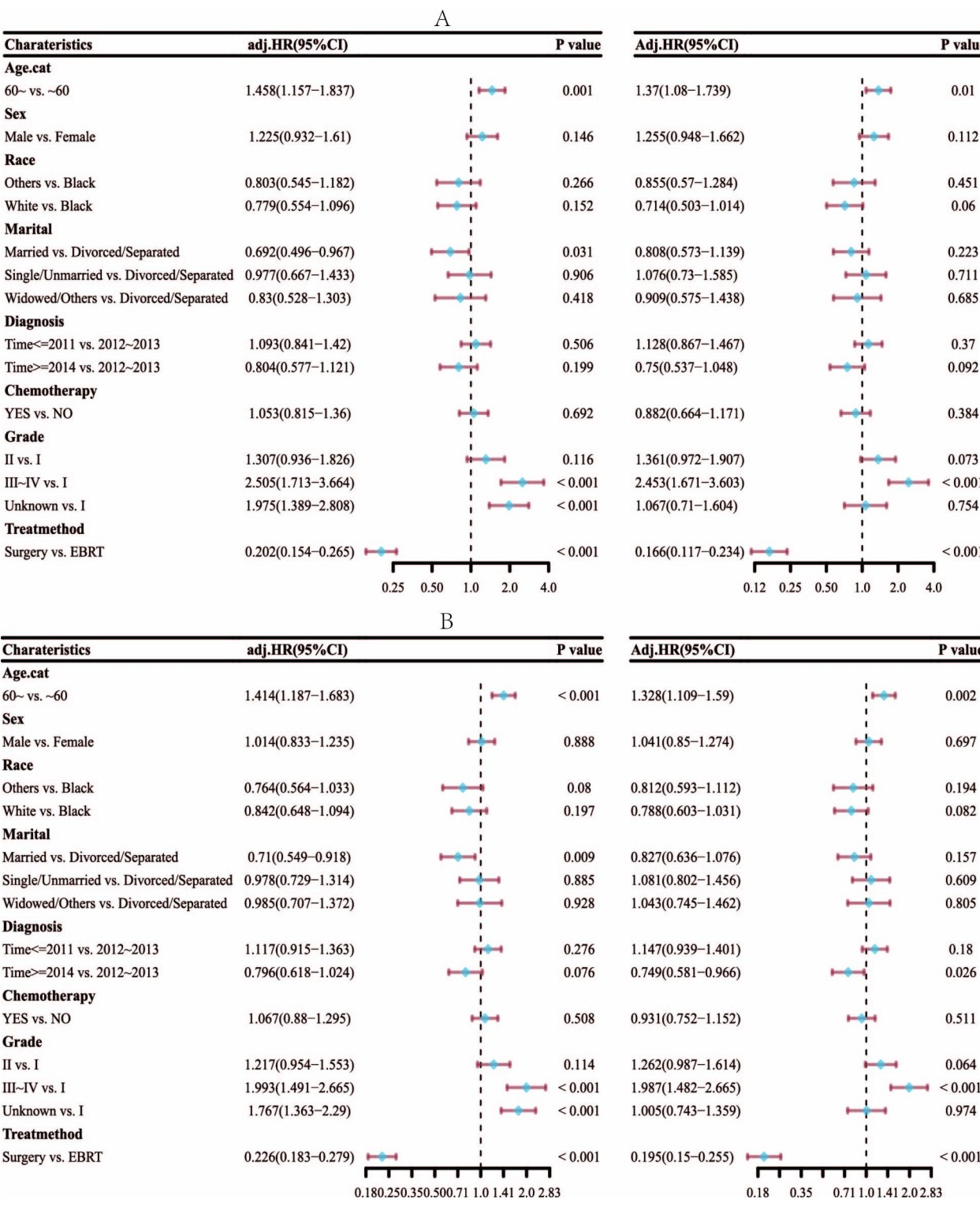

**Fig 2.** Multivariate Cox analysis of DSS (A) and OS (B).

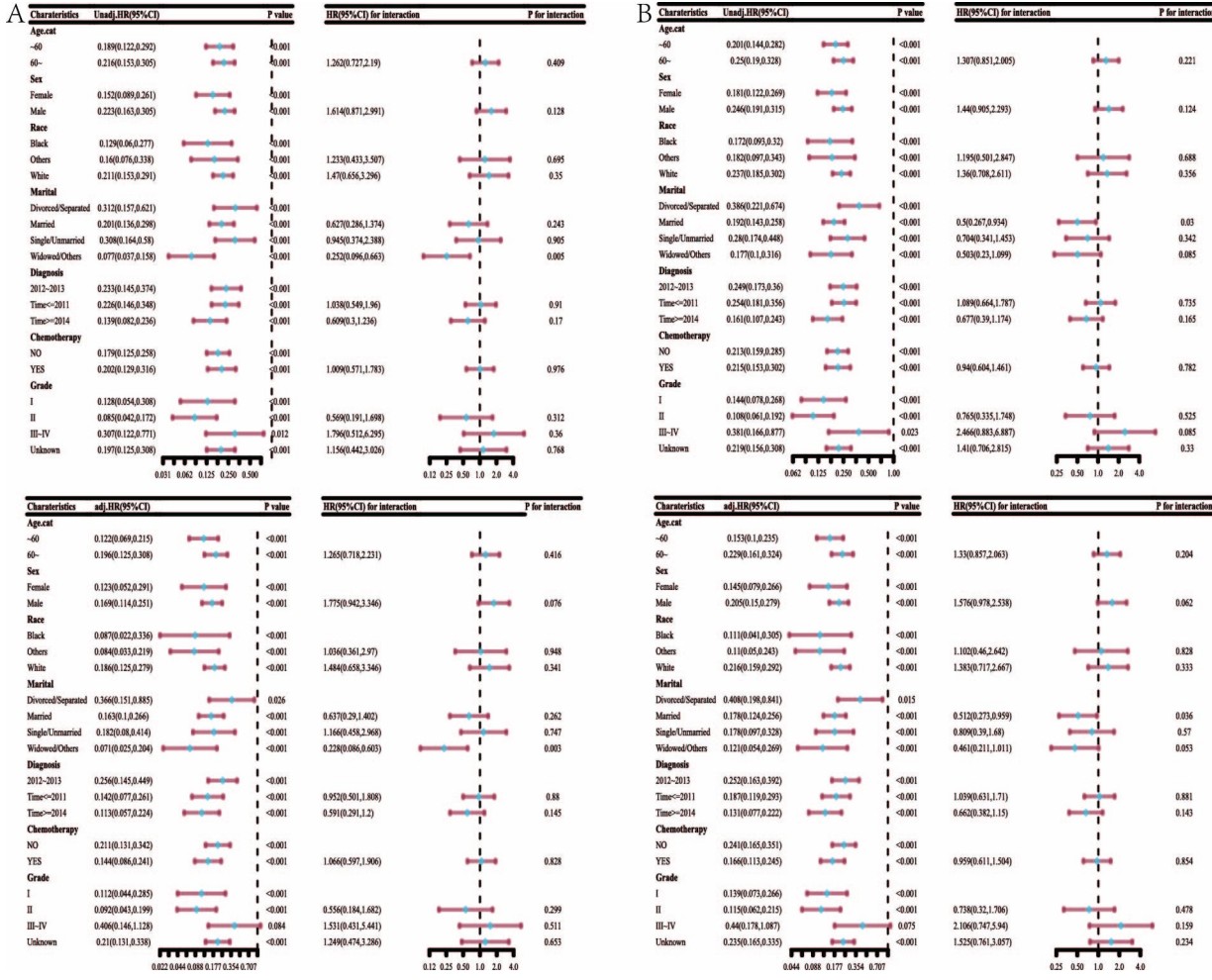

**Fig 3.** Subgroup analysis and interaction test of DSS (A) and OS (B).

was higher in those who were < 60 years old, white, widowed/other, females, diagnosed during or after 2014, and underwent chemotherapy compared with that in other groups.

Fig 9A shows the given conditional survival of OS and DSS for patients who underwent surgery. For OS, the survival probability to live 0–6 years was 100%, 94%, 87%, 81%, 81%, 74%, and 71%, respectively. For DSS, the survival probability to live 0–6 years was 100%, 96%, 93%, 87%, 87%, 83%, and 80%, respectively. Fig 9B shows the given conditional survival of OS and DSS for patients who underwent EBRT. For OS, the survival probability to live 0–6 years was 100%, 78%, 56%, 34%, 28%, 28%, and 21%, respectively. For DSS, the survival probability to live 0–6 years was 100%, 84%, 69%, 52%, 49%, 49%, and 42%, respectively.

## Discussion

The main treatment approach for early-stage liver cancer patients is liver cancer resection. Patients with early, small-sized liver tumor who either reject or cannot be operated on can be treated with local external radiation [7]. Liver tumors are the second most sensitive tumors to radiotherapy, with moderate to high radiosensitivity, after tumor tissues that have extremely high radiosensitivity to bone marrow and lymphatic tissues [8]. EBRT mainly includes stereotactic body radiotherapy (SBRT), three-dimensional conformal radiotherapy (3DCRT),

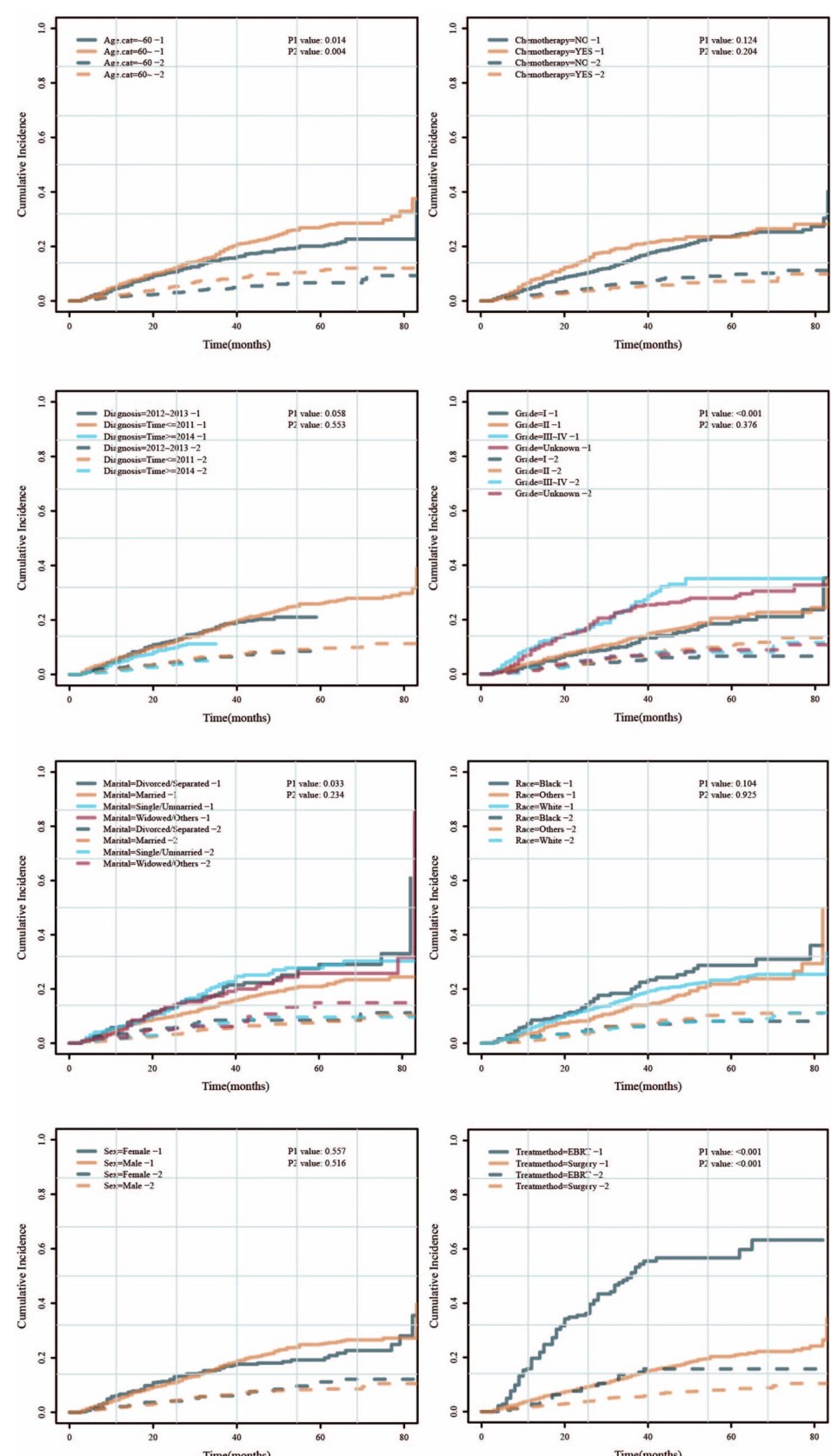

**Fig 4. Cumulative risk curve & multivariate analysis of competitive risk model.**

| Charateristics | adj.SHR(95%CI) | | P value | Adj.SHR(95%CI) | | P value |
|---|---|---|---|---|---|---|
| **Age.cat** | | | | | | |
| 60~ vs. ~60 | 1.286(1.049−1.576) | | 0.016 | 1.241(1.002−1.537) | | 0.048 |
| **Sex** | | | | | | |
| Male vs. Female | 1.074(0.85−1.358) | | 0.55 | 1.07(0.837−1.367) | | 0.59 |
| **Race** | | | | | | |
| Others vs. Black | 0.684(0.484−0.968) | | 0.032 | 0.724(0.498−1.051) | | 0.089 |
| White vs. Black | 0.794(0.59−1.069) | | 0.13 | 0.733(0.534−1.006) | | 0.054 |
| **Marital** | | | | | | |
| Married vs. Divorced/Separated | 0.73(0.542−0.983) | | 0.038 | 0.853(0.621−1.173) | | 0.33 |
| Single/Unmarried vs. Divorced/Separated | 1.008(0.717−1.417) | | 0.96 | 1.093(0.769−1.555) | | 0.62 |
| Widowed/Others vs. Divorced/Separated | 0.923(0.625−1.364) | | 0.69 | 1.019(0.688−1.508) | | 0.93 |
| **Diagnosis** | | | | | | |
| Time<=2011 vs. 2012~2013 | 1.161(0.925−1.457) | | 0.2 | 1.205(0.958−1.515) | | 0.11 |
| Time>=2014 vs. 2012~2013 | 0.781(0.582−1.049) | | 0.1 | 0.719(0.533−0.97) | | 0.031 |
| **Chemotherapy** | | | | | | |
| YES vs. NO | 1.192(0.953−1.491) | | 0.12 | 1.016(0.783−1.319) | | 0.9 |
| **Grade** | | | | | | |
| II vs. I | 1.092(0.82−1.453) | | 0.55 | 1.158(0.866−1.546) | | 0.32 |
| III~IV vs. I | 2.117(1.521−2.945) | | < 0.001 | 2.168(1.544−3.045) | | < 0.001 |
| Unknown vs. I | 1.841(1.366−2.482) | | < 0.001 | 1.029(0.713−1.486) | | 0.88 |
| **Treatmethod** | | | | | | |
| Surgery vs. EBRT | 0.227(0.178−0.288) | | < 0.001 | 0.206(0.149−0.284) | | < 0.001 |
| | | 0.18  0.35  0.71 1.0 1.41 2.0 2.83 | | | 0.25  0.50  1.0  2.0 | |

**Fig 5. Multivariate analysis of competitive risk model.**

intensity modulated radiotherapy (IMRT), and volume modulated arc radiotherapy (VMAT). Protons and carbon ions are also used to treat patients with liver cancer, but are more expensive [9]. SBRT is the most successful approach for the treatment of small-sized liver cancer, with a main focus on early HCC [10]. For lesions of all sizes, SBRT showed a relatively high 1-year OS and a low incidence of acute grade 3+ complications in HCC [11]. The significance of SBRT in HCC treatment has always been proven, regardless of the size of the study population in well-designed phase II trials, being either a relatively small retrospective cohort or a large series [12]. Currently, radiation therapy has become an indispensable part of the comprehensive treatment of liver cancer [13]. In the guidelines of some countries such as South Korea [14], radiotherapy is listed as the priority treatment for patients with early, middle, and late stage HCC. However, it is used as a treatment alternative when other standard treatments in some guidelines, such as NCCN (National Committee on Computer Network), are not feasible [15]. Our results (78% 1-year OS after external beam radiation) are close to the 1-year overall survival rate of 73.6–81.1% reported in previous studies for the treatment of early-stage liver cancer. This indicates that EBRT has a good therapeutic effect on early-stage liver cancer [16]. Therefore, the aim of this study was to further evaluate the prognosis of patients with early-stage liver cancer undergoing either surgery or external radiation, to guide clinical decision-making.

The competitive risk model established in this study suggests that age, disease, time of diagnosis, tumor tissue grade, and treatment are all independent factors for the prognosis of liver cancer patients. This study also reported that patients < 60 years old, with delayed diagnosis, low histological grade, and who had undergone surgical treatment showed good prognosis. Surgery showed a higher survival rate than EBRT (HR = 0.224, 95% CI [0.139–0.36], $p$ <0.001). After PSM, subgroup analysis and interaction test found that all patients can benefit from surgery and external radiation. The benefit of chemotherapy in female, white, widowed/other, <60 years old patients, and those with a diagnosis time during or after 2014 is greater than that of other groups.

Our study reports that surgery shows a higher survival rate among HCC patients than EBRT. After PSM, subgroup analysis and interaction test found that all HCC patients can benefit from surgery and external radiation. The group of white, widowed/other, female, < 60

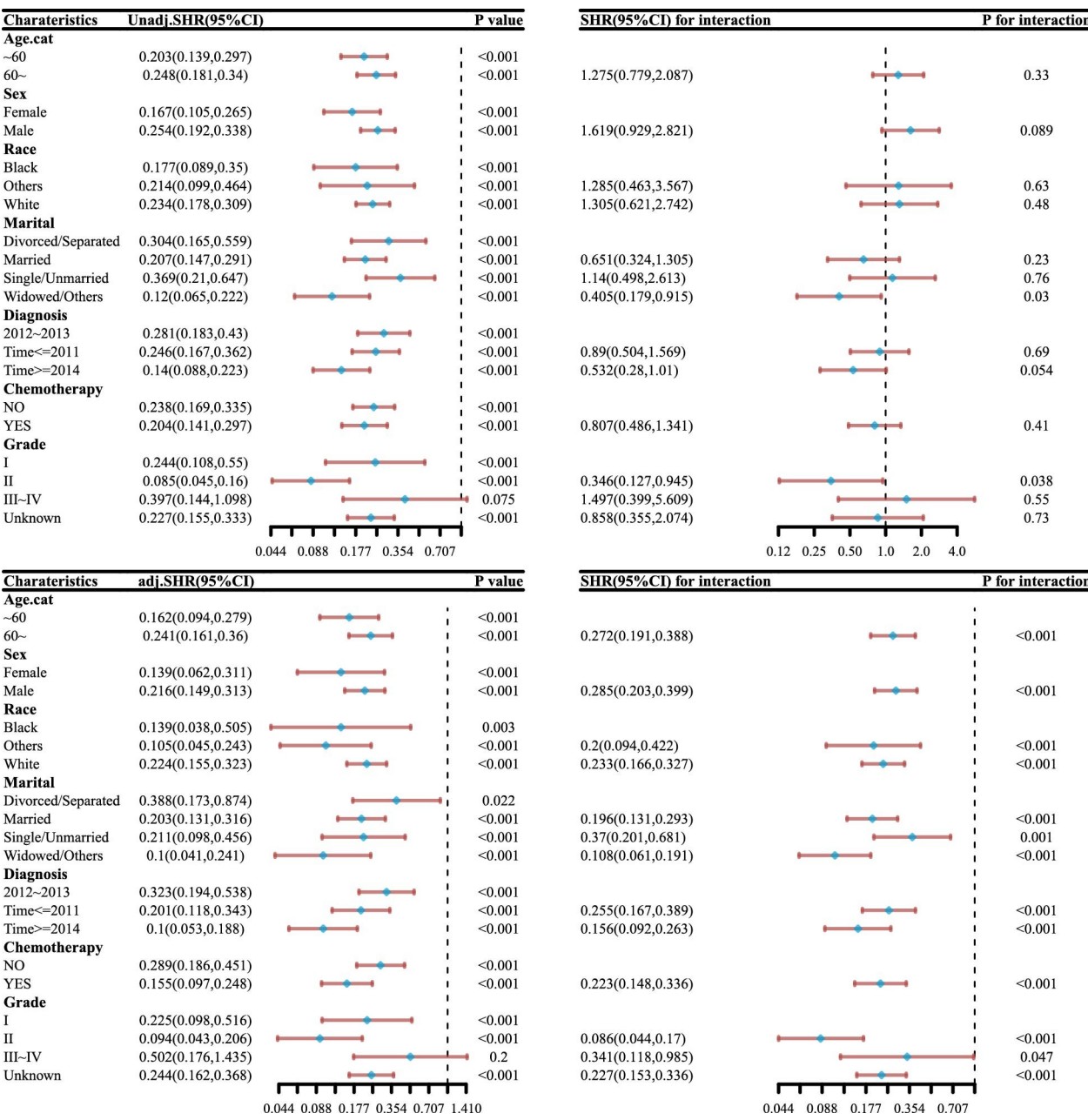

**Fig 6. Subgroup analysis and interaction test of competitive risk model.**

years old patients, with a diagnosis time during or after 2014, and receiving chemotherapy showed a greater benefit than that of other groups.

Through the conditional survival rate, we found that as the survival time of liver cancer patients increased, the survival curve showed a slower decline in surgery than that of radiotherapy patients, and stabilized after about 3 years. The survival curve of radiotherapy patients dropped significantly within 4 years, and then stabilized thereafter.

The propensity ratio scoring method is one of the commonly used methods to control confounding factors in real-world research. Its basic principle is to express the influence of multiple confounding factors in a comprehensive propensity score to correct the imbalance of data

**Table 2. Baseline data table before and after PSM matching.**

| | Unmatched | | | | | | |
|---|---|---|---|---|---|---|---|
| | level | Overall | EBRT | Surgery | p | test | SMD |
| n | | 2155 | 183 | 1972 | | | |
| Age. cat, n (%) | ~60 | 1004 (46.6) | 70 (38.3) | 934 (47.4) | 0.022 | | 0.185 |
| | 60~ | 1151 (53.4) | 113 (61.7) | 1038 (52.6) | | | |
| Sex, n (%) | Female | 581 (27.0) | 48 (26.2) | 533 (27.0) | 0.884 | | 0.018 |
| | Male | 1574 (73.0) | 135 (73.8) | 1439 (73.0) | | | |
| Race, n (%) | Black | 251 (11.6) | 17 (9.3) | 234 (11.9) | <0.001 | | 0.437 |
| | Others | 511 (23.7) | 18 (9.8) | 593 (25.0) | | | |
| | White | 1393 (64.6) | 148 (80.9) | 1245 (63.1) | | | |
| Marital, n (%) | Divorced/Separated | 261 (12.1) | 32 (17.5) | 229 (11.6) | 0.039 | | 0.215 |
| | Married | 1269 (58.9) | 92 (50.3) | 1177 (59.7) | | | |
| | Single/Unmarried | 392 (18.2) | 35 (19.1) | 357 (18.1) | | | |
| | Widowed/Others | 233 (10.8) | 24 (13.1) | 209 (10.6) | | | |
| Diagnosis, n (%) | 2012~2013 | 667 (31.0) | 53 (29.0) | 614 (31.1) | 0.812 | | 0.05 |
| | Time< = 2011 | 631 (29.3) | 54 (29.5) | 577 (29.3) | | | |
| | Time> = 2014 | 857 (39.8) | 76 (41.5) | 781 (39.6) | | | |
| Chemotherapy, n (%) | NO | 1610 (74.7) | 90 (49.2) | 1520 (77.1) | <0.001 | | 0.604 |
| | YES | 545 (25.3) | 93 (50.8) | 452 (22.9) | | | |
| Grade, n (%) | I | 511 (23.7) | 18 (9.8) | 493 (25.0) | <0.001 | | 1.482 |
| | II | 914 (42.4) | 17 (9.3) | 897 (45.5) | | | |
| | III~IV | 272 (12.6) | 11 (6.0) | 261 (13.2) | | | |
| | Unknown | 458 (21.3) | 137 (74.9) | 321 (16.3) | | | |
| | Matched | | | | | | |
| | Level | Overall | EBRT | Surgery | p | test | SMD |
| n | | 362 | 181 | 181 | | | |
| Age. cat, n (%) | ~60 | 159 (43.9) | 70 (38.3) | 89 (49.2) | 0.057 | | 0.213 |
| | 60~ | 203 (56.1) | 111 (61.3) | 92 (50.8) | | | |
| Sex, n (%) | Female | 119 (32.9) | 47 (26.0) | 72 (39.8) | 0.007 | | 0.297 |
| | Male | 243 (67.1) | 134 (74.0) | 109 (60.2) | | | |
| Race, n (%) | Black | 45 (12.4) | 17 (9.4) | 28 (15.5) | 0.011 | | 0.318 |
| | Others | 50 (13.8) | 18 (9.9) | 32 (17.7) | | | |
| | White | 267 (73.8) | 146 (80.7) | 121 (66.9) | | | |
| Marital, n (%) | Divorced/Separated | 69 (19.1) | 31 (17.1) | 38 (21.0) | 0.126 | | 0.253 |
| | Married | 171 (47.2) | 92 (50.8) | 79 (43.6) | | | |
| | Single/Unmarried | 84 (23.2) | 35 (19.3) | 49 (27.1) | | | |
| | Widowed/Others | 38 (10.5) | 23 (12.7) | 15 (8.3) | | | |
| Diagnosis, n (%) | 2012~2013 | 110 (30.4) | 53 (29.3) | 57 (31.5) | 0.88 | | 0.051 |
| | Time< = 2011 | 105 (29.0) | 54 (29.8) | 51 (28.2) | | | |
| | Time> = 2014 | 147 (40.6) | 74 (40.9) | 73 (40.3) | | | |
| Chemotherapy, n (%) | NO | 167 (46.1) | 90 (49.7) | 77 (42.5) | 0.206 | | 0.144 |
| | YES | 195 (53.9) | 91 (50.3) | 104 (57.5) | | | |
| Grade, n (%) | I | 31 (8.6) | 18 (9.9) | 13 (7.2) | 0.625 | | 0.14 |
| | II | 40 (11.0) | 17 (9.4) | 23 (12.7) | | | |
| | III~IV | 21 (5.8) | 11 (6.1) | 10 (5.5) | | | |
| | Unknown | 270 (74.6) | 135 (74.6) | 135 (74.6) | | | |

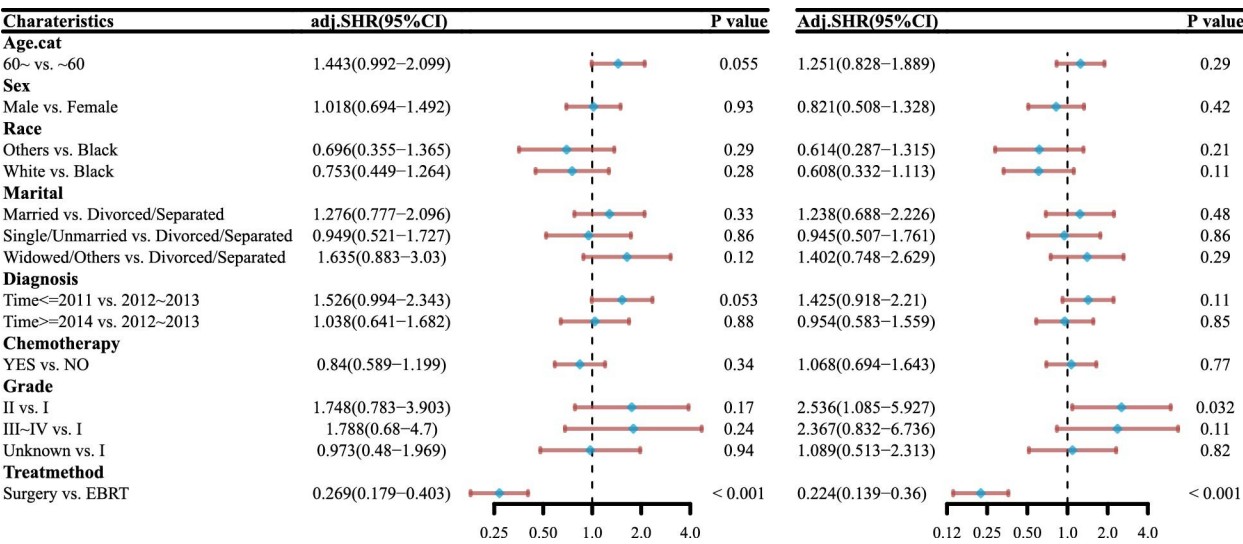

| Charateristics | adj.SHR(95%CI) | | P value | Adj.SHR(95%CI) | | P value |
|---|---|---|---|---|---|---|
| **Age.cat** | | | | | | |
| 60~ vs. ~60 | 1.443(0.992−2.099) | | 0.055 | 1.251(0.828−1.889) | | 0.29 |
| **Sex** | | | | | | |
| Male vs. Female | 1.018(0.694−1.492) | | 0.93 | 0.821(0.508−1.328) | | 0.42 |
| **Race** | | | | | | |
| Others vs. Black | 0.696(0.355−1.365) | | 0.29 | 0.614(0.287−1.315) | | 0.21 |
| White vs. Black | 0.753(0.449−1.264) | | 0.28 | 0.608(0.332−1.113) | | 0.11 |
| **Marital** | | | | | | |
| Married vs. Divorced/Separated | 1.276(0.777−2.096) | | 0.33 | 1.238(0.688−2.226) | | 0.48 |
| Single/Unmarried vs. Divorced/Separated | 0.949(0.521−1.727) | | 0.86 | 0.945(0.507−1.761) | | 0.86 |
| Widowed/Others vs. Divorced/Separated | 1.635(0.883−3.03) | | 0.12 | 1.402(0.748−2.629) | | 0.29 |
| **Diagnosis** | | | | | | |
| Time<=2011 vs. 2012~2013 | 1.526(0.994−2.343) | | 0.053 | 1.425(0.918−2.21) | | 0.11 |
| Time>=2014 vs. 2012~2013 | 1.038(0.641−1.682) | | 0.88 | 0.954(0.583−1.559) | | 0.85 |
| **Chemotherapy** | | | | | | |
| YES vs. NO | 0.84(0.589−1.199) | | 0.34 | 1.068(0.694−1.643) | | 0.77 |
| **Grade** | | | | | | |
| II vs. I | 1.748(0.783−3.903) | | 0.17 | 2.536(1.085−5.927) | | 0.032 |
| III~IV vs. I | 1.788(0.68−4.7) | | 0.24 | 2.367(0.832−6.736) | | 0.11 |
| Unknown vs. I | 0.973(0.48−1.969) | | 0.94 | 1.089(0.513−2.313) | | 0.82 |
| **Treatmethod** | | | | | | |
| Surgery vs. EBRT | 0.269(0.179−0.403) | | < 0.001 | 0.224(0.139−0.36) | | < 0.001 |

**Fig 7. Multi-factor forest diagram of competitive risk model after PSM.**

between groups [17]. Previous studies have shown that significant risk factors for HCC recurrence mainly include HCC lesion size (especially >3 cm), etiology, serum albumin levels, and serum alpha-fetoprotein levels [18]. Factors affecting survival include patient's age and Child Pugh staging. The patient's age has also been considered to be related to the incidence of the disease and post-treatment complications [19]. Our study was a retrospective large-sample study, and used PSM to enhance credibility and reduce potential confounding factors and selection bias between the case group and the control group. Our study findings conformed with those of the previous studies, where PSM, subgroup analysis and interaction test of competitive risk model showed that the treatment benefit was higher in the group < 60 years old, female, widowed/others, white patients who underwent chemotherapy and were diagnosed during or after 2014.

By comparing the results of the traditional survival analysis COX model and the competitive risk model, a previous study [20]showed that the use of the COX model could not provide an accurate estimate of the impact value, because it only considered the results of a single factor, and thus might overestimate or underestimate the impact of the independent risk factors. If the proportion of competition events was >10%, using traditional methods could cause serious bias. On the other hand, a proportion of competition events <10% may have false positives or false negatives [21]. For survival analysis, competitive risk model can divide the end points of survival data into multiple categories and eliminate the impact of competitive events on prognosis research [22]; thus it is considered a more effective survival analysis model. Previous studies mostly used traditional survival analysis methods, ignoring the existence of competing events, and hence, the risk of death from cancer may be overestimated [23]. This study was a multi-center, large sample study of patients with stage I liver cancer based on the SEER database. The study included a total of 2155 cases and had a strong statistical power, which compensated for the shortcomings of the small sample size studies of general clinical research, and had high clinical reference value. In this study, 523 people died in the OS analysis, of which 300 people died in the DSS analysis, accounting for 57.36% of total deaths. Therefore, the proportion of deaths due to other competitive events was 42.64%, which was suitable for analysis using a competitive risk model to incorporate multiple factors, as shown in this study. The cumulative risk curve showed that there were significant differences between the two groups at different ages and with different

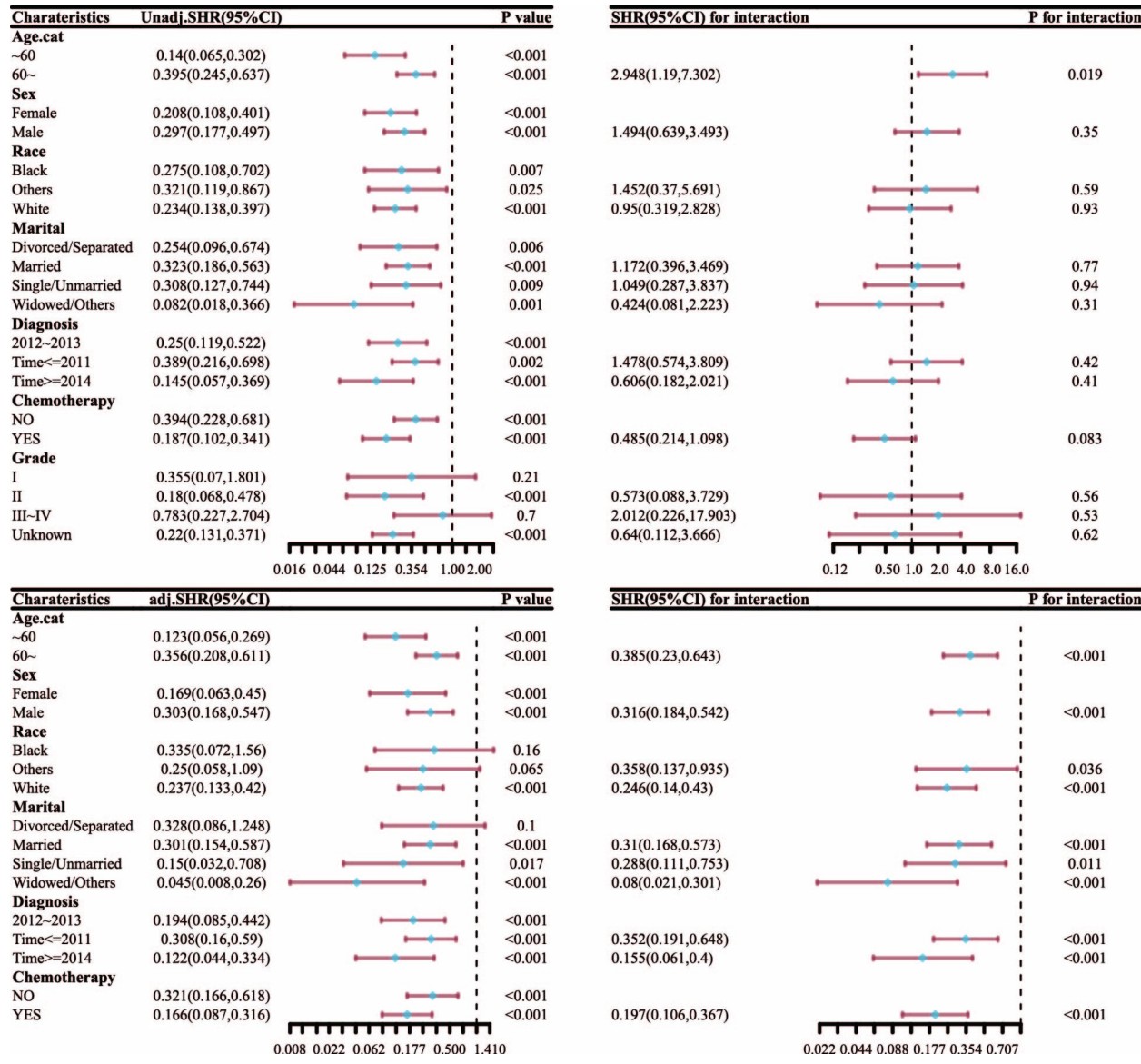

**Fig 8. Subgroup analysis and interaction test of competitive risk model after PSM.**

treatments. However, there was no significant difference between the groups with respect to receiving chemotherapy, having different time of diagnosis, or belonging to different races and sexes, which further supported the superiority of our usage of the competitive risk model.

Generally, the survival rate post liver cancer resection is evaluated based on the date of surgery. However, this traditional survival curve may not provide an accurate prediction of long-term survival, mainly because the recurrence and mortality rates are generally the highest in the first few years after surgery. Survival rate is dynamic and is directly related to the duration of time between the beginning of the treatment course and the time of evaluation [24]. Estimates of survival time vary over time, thus conditional survival evaluation is a more meaningful way to evaluate long-term prognosis. It is calculated based on the patient's survival time and is more relevant to the actual clinical practice because it can measure the patient's survival risk over a certain period. Conditional probability analysis is increasingly used to analyze the

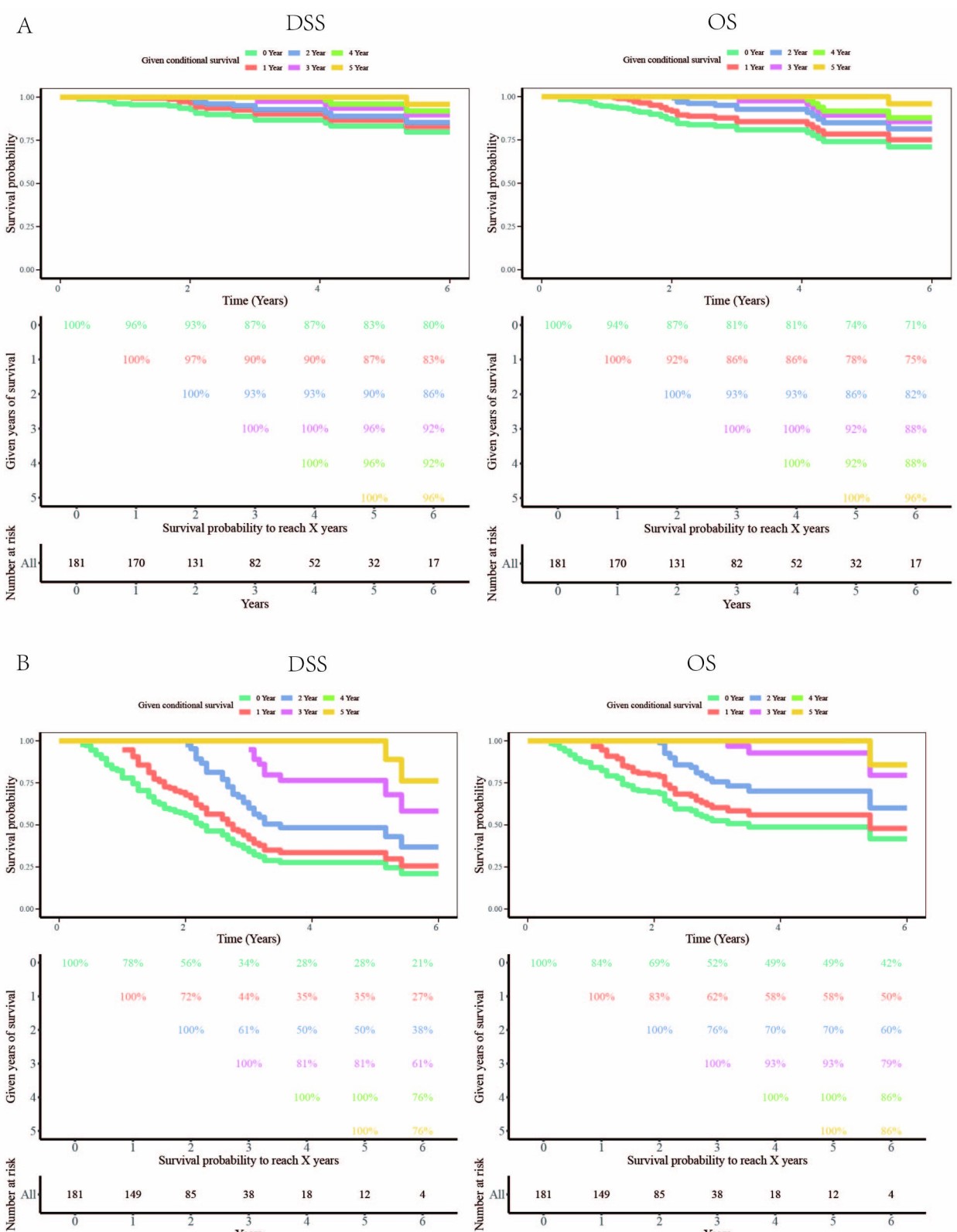

**Fig 9.** Conditional survival of DSS and OS for patients underwent surgery (A) and EBRT (B).

long-term effects of tumor characteristics [25]and is particularly suitable for comparing the difference between immediate and late-stage survival benefits after treatment [26].

In this study, we evaluated DSS and OS in early-stage liver patients after surgery or external irradiation based on a large, multi-center database. The results showed that after 3 years of surgery or 4 years of external radiation, the decline in patients' DSS and OS tends to be more stable as the survival time increases, which means that generally long-term cancer patients have better prognosis than newly diagnosed patients. This was similar to other studies [27], which showed that the impact of tumor-related factors on the long-term survival of liver cancer patients was diminished starting from the third year post surgery. This also showed that patients with early-stage liver cancer had the most survival-related events such as recurrence and metastasis within 3 years post-surgery or 4 years post external radiation. Close follow-up and re-examination are needed for timely treatment to reduce mortality. Previous studies [26] had shown that approximately 7% of patients undergoing surgical treatment for HCC with mild gross vascular infiltration were expected to be cured, and conditional survival probability analysis could also be used to guide the follow-up of such patients after surgery [15]. These findings were different from our study because their research object was liver cancer patients with mild gross vascular infiltration who showed a worse prognosis than that of stage I liver cancer patients. In addition, the number of cases in their study was limited to only a few hundred.

The current HCC NCCN guidelines recommend that all liver cancer patients should be followed up every 3–6 months for the first 2 years after surgery, but there is no optimal postoperative/post-radiotherapy follow-up strategy for patients with early-stage liver cancer. Conditional survival evaluation can provide more valuable information for the determination of subsequent strategies. On the other hand, DSS survival rate showed a slower decrease than OS after surgery and external radiation, indicating that other factors that affect the survival of liver cancer patients 3 years after surgery or 4 years after external radiation have become more important. Therefore, this study not only compares the difference in survival of patients with early-stage liver cancer after surgery and external radiation and the relationship between clinicopathological characteristics and prognosis, but also conducts for the first time a dynamic analysis of the survival of patients with early-stage liver cancer.

One of the limitations of this study is that most of the data collected by the SEER database were clinical indicators, excluding laboratory examinations, imaging examinations and others, thus the accuracy of the clinical TNM staging of patients without surgery needed to be improved. Second, there were some missing clinical indicators in the SEER database. Third, this study only conducted internal verification of the prognostic model, and we hoped to conduct further external verification to evaluate the applicability of the predictive model in the population in a more comprehensive manner. All of these shortcomings might affect the accuracy of the prognostic prediction model constructed in this study. In the future, more clinical studies are needed for further verification of the results.

In summary, this study is the first to analyze and study the prognosis model of early-stage liver cancer patients established in the SEER database based on the competitive risk model and conditional survival analysis, which introduces new ideas and methods for the study of predictive models. It can more comprehensively and accurately predict the prognosis of patients with early-stage liver cancer, provide important parameters for the prevention of tumor recurrence, and provide personalized treatment plans and prognostic judgments.

## Supporting information

**S1 Checklist. STROBE statement—checklist of items that should be included in reports of observational studies.**
(DOC)

**S1 Fig. PH test of each variable for DSS.**
(JPG)

**S2 Fig. PH test of each variable for OS.**
(JPG)

**S3 Fig. Balance test before and after PSM.**
(JPG)

## Acknowledgments

We are grateful for the support of Jiangsu Provincial Health Commission key project (K2023005). We also thank the Jiangsu Provincial Medical Youth Talent (QNRC2016816), the Project of Jiangsu Provincial Health and Family Planning Commission (H2018090).

## Author Contributions

**Conceptualization:** Rong Chen.

**Data curation:** Rong Chen.

**Formal analysis:** Muhao Xu.

**Investigation:** Yanli An.

**Methodology:** Muhao Xu.

**Project administration:** Yanli An.

**Resources:** Yanli An.

**Supervision:** Rong Chen.

**Writing – original draft:** Rong Chen.

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
