## [Decision Letter · Decision Letter 0]

17 Apr 2023

PONE-D-23-01992Prognostic Difference Between Surgery and External Radiation in Patients with Stage I Liver Cancer Based on Competitive Risk Model and Conditional Survival RatePLOS ONE

Dear Dr. Chen,

Thank you for submitting your manuscript to PLOS ONE. After careful consideration, we feel that it has merit but does not fully meet PLOS ONE’s publication criteria as it currently stands. Therefore, we invite you to submit a revised version of the manuscript that addresses the points raised during the review process.

We look forward to receiving your revised manuscript.

Kind regards,

Calogero Casà

Academic Editor

PLOS ONE

Journal Requirements:

“National Natural Science Foundation of China (81827805), National Key R&D Program of China (2018YFA0704100, 2018YFA0704104)”

Reviewers' comments:

Reviewer's Responses to Questions

**Comments to the Author**

1. Is the manuscript technically sound, and do the data support the conclusions?

Reviewer #1: Yes

Reviewer #2: Partly

2. Has the statistical analysis been performed appropriately and rigorously? 

Reviewer #1: Yes

Reviewer #2: Yes

3. Have the authors made all data underlying the findings in their manuscript fully available?

Reviewer #1: Yes

Reviewer #2: No

4. Is the manuscript presented in an intelligible fashion and written in standard English?

Reviewer #1: Yes

Reviewer #2: Yes

5. Review Comments to the Author

Reviewer #1: it might be helpful for the clinician to specify which radiotherapy treatment schedules to propose for patients with early stage hepatocarcinoma. This is useful to guide the surgeon and radiation oncologist on which treatment schedules and techniques have achieved better results.

Reviewer #2: The authors performed an analysis of a large number of patients with liver cancer from SEER dataset assessing the role of surgery and radiotherapy in this subset.

The theme is of interest, but the analysis should consider some hints that could relate to actual treatment options.

Major revisions:

I suggest removing the marital status in the description of results.

The authors should add information about the surgical approach; regarding EBRT they should refer to the total dose and the delivery technique (3DCRT or VMAT or SBRT or Particle therapy). Regarding these variables, they should consider their impact and give some hints in the discussion section.

When considering the different time interval for surgery and the related different outcomes, the authors should provide considerations about it (different techniques?).

Also, some language tips:

- Line 69: replace “fatality” with “mortality”.

- Line 69: remove “management”.

- Line 72: replace “management” with “treatment”.

- Line 92: please provide definition of “DSS”.

6. PLOS authors have the option to publish the peer review history of their article (what does this mean?). If published, this will include your full peer review and any attached files.

Reviewer #1: No

Reviewer #2: No

---

## [Author Response · Author response to Decision Letter 0]

15 Sep 2023

Reply：Thanks for the editor's suggestion, we have checked that our manuscript meet the requirements of PLOS ONE.

“National Natural Science Foundation of China (81827805), National Key R&D Program of China (2018YFA0704100, 2018YFA0704104)”

Reply：Thanks for the editor's suggestion. We have added instructions to the Fund section as requested.

Reply：Thanks to the editor's suggestion, we have added a description of data acquisition in the corresponding section as“Raw data are made available on SEER(https://seer.cancer.gov/) database”.

Reply：Thanks to the editor's suggestion, Raw data are made available on SEER (https://seer.cancer.gov/) database.

Reply：Thanks to the editor's suggestion, I have provided the ORCID iD.

Reply：Thanks to the editor's suggestion, I have made corresponding modifications.



Reviewers' comments:

Reviewer's Responses to Questions

Comments to the Author

1. Is the manuscript technically sound, and do the data support the conclusions?

Reviewer #1: Yes

Reviewer #2: Partly

2. Has the statistical analysis been performed appropriately and rigorously?

Reviewer #1: Yes

Reviewer #2: Yes

3. Have the authors made all data underlying the findings in their manuscript fully available?

Reviewer #1: Yes

Reviewer #2: No

4. Is the manuscript presented in an intelligible fashion and written in standard English?

Reviewer #1: Yes

Reviewer #2: Yes

5. Review Comments to the Author

Reviewer #1: it might be helpful for the clinician to specify which radiotherapy treatment schedules to propose for patients with early stage hepatocarcinoma. This is useful to guide the surgeon and radiation oncologist on which treatment schedules and techniques have achieved better results.

Reviewer #2: The authors performed an analysis of a large number of patients with liver cancer from SEER dataset assessing the role of surgery and radiotherapy in this subset.

The theme is of interest, but the analysis should consider some hints that could relate to actual treatment options.

Major revisions:

I suggest removing the marital status in the description of results.

Reply：Thanks for the reviewer's suggestion. Previous studies have found that marital status may have an impact on the incidence of liver cancer (PMID: 34774036, PMID: 33428602), so we included marital status in the analysis. At the same time, we also carried out a new analysis after removing the variable of marital status. In the multi-factor regression analysis, the main results and conclusions of this study were not affected without correcting marital status, that is Surgery was better than EBRT for patients with stage I liver cancer (as shown in the figure below). This not only proves the robustness of our conclusions, but also provides a basis for including marital status in our analysis. 

The authors should add information about the surgical approach; regarding EBRT they should refer to the total dose and the delivery technique (3DCRT or VMAT or SBRT or Particle therapy). Regarding these variables, they should consider their impact and give some hints in the discussion section.

When considering the different time interval for surgery and the related different outcomes, the authors should provide considerations about it (different techniques?).

Reply：Thanks for the reviewer's suggestion. Based on the SEER database, this study found whether surgery and EBRT were disease-specific death factors in Patients with Stage I Liver Cancer through competitive risk model and correction of bias caused by competitive events, and found that the survival rate of surgery was higher than that of EBRT. Unfortunately, SEER did not record the specific information of surgery entry path, time and EBRT dose. Therefore, for patients who also received treatment, the impact of treatment interval and specific treatment methods on patient prognosis has not been studied. However, these variables are worth studying, so we have discussed them in the discussion section. Meanwhile, we are also conducting a prospective cohort study, which cannot be completed in a short time due to follow-up and other work in the cohort study. We are working hard to promote the above work and hope to carry out a series of research results in the future.

Also, some language tips:

- Line 69: replace “fatality” with “mortality”.

- Line 69: remove “management”.

- Line 72: replace “management” with “treatment”.

- Line 92: please provide definition of “DSS”.

Reply：Thanks for the reviewer's suggestion. I have made modifications according to the above requirements.

---

## [Decision Letter · Decision Letter 1]

17 Jan 2024

Prognostic Difference Between Surgery and External Radiation in Patients with Stage I Liver Cancer Based on Competitive Risk Model and Conditional Survival Rate

PONE-D-23-01992R1

Dear Dr. Chen,

Good morning. We’re pleased to inform you that your manuscript has been judged scientifically suitable for publication and will be formally accepted for publication once it meets all outstanding technical requirements.

Kind regards,

Calogero Casà

Academic Editor

PLOS ONE

Additional Editor Comments (optional):

all comments were considered and the text was appropriately modified

Reviewers' comments:

Reviewer's Responses to Questions

**Comments to the Author**

1. If the authors have adequately addressed your comments raised in a previous round of review and you feel that this manuscript is now acceptable for publication, you may indicate that here to bypass the “Comments to the Author” section, enter your conflict of interest statement in the “Confidential to Editor” section, and submit your "Accept" recommendation.

Reviewer #2: All comments have been addressed

Reviewer #3: (No Response)

2. Is the manuscript technically sound, and do the data support the conclusions?

Reviewer #2: Yes

Reviewer #3: Yes

3. Has the statistical analysis been performed appropriately and rigorously? 

Reviewer #2: N/A

Reviewer #3: N/A

4. Have the authors made all data underlying the findings in their manuscript fully available?

Reviewer #2: No

Reviewer #3: Yes

5. Is the manuscript presented in an intelligible fashion and written in standard English?

Reviewer #2: Yes

Reviewer #3: Yes

6. Review Comments to the Author

Reviewer #2: The authors have provided the requested revision to the manuscript. The paper is suitable for acceptance.

Reviewer #3: the only possible issue relates to the comparison of a very different number of patients in terms of size, 1972 for surgery and and 183 for EBRT; therefore, the comparison may not be totally reliable

7. PLOS authors have the option to publish the peer review history of their article (what does this mean?). If published, this will include your full peer review and any attached files.

Reviewer #2: No

Reviewer #3: No

---

## [Editor Report · Acceptance letter]

19 Mar 2024

PONE-D-23-01992R1 

PLOS ONE

Dear Dr. Chen, 

I'm pleased to inform you that your manuscript has been deemed suitable for publication in PLOS ONE. Congratulations! Your manuscript is now being handed over to our production team.

Kind regards, 

on behalf of

Dr. Calogero Casà 

Academic Editor

PLOS ONE